# Control of TSC2-Rheb signaling axis by arginine regulates mTORC1 activity

Bernadette Carroll[1]*, Dorothea Maetzel[2], Oliver DK Maddocks[3], Gisela Otten[1], Matthew Ratcliff[1], Graham R Smith[1], Elaine A Dunlop[4], João F Passos[1], Owen R Davies[1], Rudolf Jaenisch[2], Andrew R Tee[4], Sovan Sarkar[5], Viktor I Korolchuk[1]*

[1]Institute for Cell and Molecular Biosciences, Newcastle University, Newcastle upon Tyne, United Kingdom; [2]Whitehead Institute for Biomedical Research, Massachusetts Institute of Technology, Cambridge, United States; [3]The Beatson Institute for Cancer Research, Glasgow, United Kingdom; [4]Institute of Cancer and Genetics, Cardiff University, Cardiff, United Kingdom; [5]Institute of Cancer and Genomic Sciences, Institute of Biomedical Research, College of Medical and Dental Sciences, University of Birmingham, Birmingham, United Kingdom

**Abstract** The mammalian target of rapamycin complex 1 (mTORC1) is the key signaling hub that regulates cellular protein homeostasis, growth, and proliferation in health and disease. As a prerequisite for activation of mTORC1 by hormones and mitogens, there first has to be an available pool of intracellular amino acids. Arginine, an amino acid essential during mammalian embryogenesis and early development is one of the key activators of mTORC1. Herein, we demonstrate that arginine acts independently of its metabolism to allow maximal activation of mTORC1 by growth factors via a mechanism that does not involve regulation of mTORC1 localization to lysosomes. Instead, arginine specifically suppresses lysosomal localization of the TSC complex and interaction with its target small GTPase protein, Rheb. By interfering with TSC-Rheb complex, arginine relieves allosteric inhibition of Rheb by TSC. Arginine cooperates with growth factor signaling which further promotes dissociation of TSC2 from lysosomes and activation of mTORC1. Arginine is the main amino acid sensed by the mTORC1 pathway in several cell types including human embryonic stem cells (hESCs). Dependence on arginine is maintained once hESCs are differentiated to fibroblasts, neurons, and hepatocytes, highlighting the fundamental importance of arginine-sensing to mTORC1 signaling. Together, our data provide evidence that different growth promoting cues cooperate to a greater extent than previously recognized to achieve tight spatial and temporal regulation of mTORC1 signaling.

*For correspondence:
bernadette.carroll@newcastle.ac.uk (BC);
viktor.korolchuk@ncl.ac.uk (VIK)

**Competing interests:** The author declares that no competing interests exist.

## Introduction

Cells rely on the ability to appropriately sense and respond to intra- and extracellular stimuli in order to ensure their proper function, maintenance, and growth. Dysregulation of pathways required to control this homeostasis frequently occur in human pathologies such as cancer, metabolic syndrome, and aging. As such, intense efforts have been made to understand the regulatory signaling pathways and underlying mechanisms via which cells sense anabolic and catabolic signals.

The mammalian target of rapamycin complex 1 (mTORC1) plays a fundamentally important role in the integration of metabolic, energy, hormonal, and nutritional stimuli to promote cellular biosynthesis and suppress the catabolic process of macroautophagy (*Laplante and Sabatini, 2012*; *Carroll et al., 2015*). The two most proximal regulators of mTORC1 are small GTPases of the Ras superfamily: Rheb and the Rag GTPases. Both are considered to be resident on the cytoplasmic surface of the lysosome, a site that has in recent years come to the very forefront of mTORC1 signal

**eLife digest** Cells need to be able to sense and respond to signals from their environment. A group (or complex) of conserved proteins called mTORC1 acts a key signaling hub that regulates cell growth and many other processes. This complex can be activated by many different signals from outside the cell. However, mTORC1 can only be activated by these signals if there is also a good supply of amino acids – which are needed to make new proteins – within the cell.

The amino acids are thought to be presented to mTORC1 on the outer surface of cellular compartments known as lysosomes. A protein called Rheb on the surface of the lysosomes activates mTORC1, while a protein complex called TSC inhibits the activity of Rheb to regulate mTORC1 activity. Previous studies have shown that some amino acids influence whether mTORC1 can be activated by affecting whether it is localized to the lysosomes or not.

Here, Carroll et al. explored how an amino acid called arginine regulates mTORC1. The experiments show that arginine is the major amino acid that influences whether mTORC1 can be activated in several different types of human cell. When cells were deprived of arginine, the activity of the complex was strongly suppressed. However, microscopy showed that arginine had no effect on whether mTORC1 was found at the lysosomes or not, which suggests that arginine might be acting in a different way to other amino acids.

Further experiments found that a lack of arginine led to an increase in the number of TSC complexes at the lysosomes. This led to the inhibition of Rheb and therefore prevented mTORC1 from being activated. Together, Carroll et al.'s findings provide evidence that the different signals that regulate mTORC1 signaling cooperate to a greater extent than previously thought. A future challenge will be to understand the molecular details of how the arginine is detected.

integration (*Sancak et al., 2010*; *Buerger et al., 2006*). The Rag GTPases facilitate signaling of amino acids to mTORC1 (*Kim et al., 2008*; *Sancak et al., 2008*), a stimulus that has long been known to be both necessary and sufficient, albeit minimally, for mTORC1 activation (*Long et al., 2005b*; *Hara et al., 1998*). The current consensus is that amino acids activate mTORC1 by promoting its localization to the cytoplasmic surface of late endosomes and lysosomes (for simplicity this mTOR-positive compartment is referred to as lysosomes) via a heterodimeric complex of Rag GTPases (*Sancak et al., 2008*; *Sancak et al., 2010*). There are four mammalian Rag GTPases that show functional redundancy, A or B dimerize with C or D and unlike most small GTPases they rely on another complex, the Ragulator, for tethering them at the membrane. The Ragulator complex further acts as a guanine nucleotide exchange factor (GEF) to activate the Rag GTPases and works in opposition to the GTPase-activating protein (GAP) complex, GATOR1 (*Bar-Peled et al., 2012*; *Bar-Peled et al., 2013*; *Panchaud et al., 2013*) to ensure the tight control of Rag GTPases and therefore mTORC1 activity.

mTORC1 localization to the lysosome is thought promote its activity by bringing it into close proximity to the membrane-associated Rheb. Rheb is the master activator of mTORC1 and, although the mechanism by which GTP-loaded Rheb stimulates mTORC1 kinase activity remains unknown (*Avruch et al., 2009*), it has been demonstrated that the nucleotide and therefore activity status of Rheb is controlled by a wide range of upstream stimuli. Potent mTORC1 regulators, for example energy status via AMPK and growth factors via PI3K and Akt converge upstream of Rheb to positively or negatively control the TSC complex (*Inoki et al., 2003b*; *Huang and Manning, 2008*). The TSC complex consists of TSC1, TBC1D7, and TSC2, where TSC2 has GAP activity that specifically inactivates Rheb (*Tee et al., 2003*; *Inoki et al., 2003a*; *Garami et al., 2003*; *Zhang et al., 2003*; *Dibble et al., 2012*). Phosphorylation of TSC2 by Akt has recently been shown to regulate localization of the TSC complex to the cytoplasm and away from the lysosome thereby preventing Rheb inactivation and promoting mTORC1 activation (*Menon et al., 2014*). The regulation of TSC2 localization to the lysosome has also been associated with the amino acid/Rag GTPase axis (*Demetriades et al., 2014*), indicating that the underlying mechanisms controlling the opposing localization of mTOR and TSC2 to the lysosome involve complex interplay between different upstream stimuli.

Leucine and glutamine are the best-studied mTORC1-regulating amino acids and their mechanisms of action have been partially elucidated (*Han et al., 2012*; *Sancak et al., 2010*; *Durán et al., 2012*; *Jewell et al., 2015*). In addition to leucine and glutamine, mTORC1 has also been shown to sense the presence of arginine (*Hara et al., 1998*; *Wang et al., 2015*; *Ban et al., 2004*), an amino acid essential during embryogenesis (*Wu et al., 2009*). It has recently been suggested that arginine activates mTORC1 via a mechanism similar to that of leucine, involving Rag GTPases where a membrane transporter SLC38A9 acts as a sensor of amino acids (*Jung et al., 2015*), including arginine (*Rebsamen et al., 2015*; *Wang et al., 2015*), on lysosomal membranes.

Here, we show that arginine has an unexpected role in activating mTORC1 via the TSC/Rheb signaling axis. Most notably, we demonstrate that arginine does not significantly affect mTORC1 localization but rather, it is required for maximal growth factor signaling by preventing TSC2 localization to the lysosome and it's interaction with Rheb. We demonstrate that TSC2 localization is modulated by arginine together with the classical TSC regulators, growth factors, to tightly control Rheb and ultimately mTORC1 activity in a spatial and temporal manner. Arginine is likely to act as an intact molecule as its metabolism is not required for mTORC1 activation but it remains to be elucidated whether the effect is direct or indirect. Finally, arginine is the only amino acid that mTORC1 activity shows sensitivity to in both undifferentiated stem cells and subsequent differentiated lineages demonstrating the fundamental importance of the mechanism described herein. This report further highlights the complex signaling interplay between different growth-promoting stimuli in regulating mTORC1 at the level of lysosomes.

## Results

### Arginine is essential for mTORC1 activity

The presence of amino acids is a prerequisite for mTORC1 activity. Leucine in particular has been well documented for its role in the regulation of mTORC1 and its growth-promoting properties. However, the contribution and mechanisms of action of other amino acids are not as well understood. To further explore amino acid signaling, we investigated the sensitivity of mTORC1 to essential and conditionally essential amino acids in a panel of cell lines. Acute deprivation of leucine, arginine, or glutamine alone was observed to significantly perturb mTORC1 activity in multiple cell lines (*Figure 1—figure supplement 1A–G* and see, among others [*Durán et al., 2012*; *Nicklin et al., 2009*; *Wang et al., 2015*; *Jewell et al., 2015*]). Deprivation of other single amino acids, including isoleucine and methionine, did not significantly affect mTORC1 activity in any cell line tested (*Figure 1—figure supplement 1A–G* and data not shown). Interestingly, mTORC1 was found to be differentially sensitive to amino acids in different cell lines. For example, sensitivity to glutamine, previously associated with leucine-dependent and -independent mechanisms of mTORC1 regulation (*Durán et al., 2012*; *Nicklin et al., 2009*; *Jewell et al., 2015*), varied significantly. While glutamine was essential for mTORC1 activation in both HeLa and HEK293T cells, it was less important in Mouse Embryonic Fibroblasts (MEFs). Similarly, removal of leucine significantly suppressed mTORC1 activity in most cell lines with an exception of HeLa and U2OS. Of particular interest, we found that mTORC1 activity is dependent on arginine in every cell line tested. Therefore, this amino acid appears to be a fundamentally important regulator of mTORC1.

Simultaneous deprivation of arginine and leucine had an additive effect on mTORC1 inhibition (*Figure 1—figure supplement 1A–F*), while at the same time they (alongside glutamine which is known to be required for the transport of leucine into the cell [*Nicklin et al., 2009*]) represent a sufficient signal to activate mTORC1 (*Figure 1—figure supplement 1H*). No one individual amino acid is sufficient to activate mTORC1, but this is unlikely to be a result of lower intracellular concentrations. Thus, for example, arginine uptake is in fact increased in cells completely starved of all amino acids compared to the cells incubated with complete amino acids, presumably due to a lack of competition for transport into the cell (*Figure 1—figure supplement 1I*). mTORC1 activity induced by a combination of arginine, leucine, and glutamine is minimal compared to that of a complete set of amino acids; however, it is not further enhanced by the addition of any other individual amino acid, such as isoleucine (*Figure 1—figure supplement 1H* and data not shown). Together, these data suggest that arginine, leucine and glutamine represent the major contributors to amino acid-

dependent mTORC1 activation. Moreover, the synergistic effect of these amino acids on mTORC1 activity rather than simply concentration-dependent effects suggests they act via multiple mechanisms.

We observe that metabolism or cellular utilization of arginine does not participate in its ability to activate mTORC1 (*Figure 1—figure supplement 2A–J*). Rather, interventions that increase intracellular levels of arginine (such as knock-down of arginyl-tRNA synthetase (RARs), L-norvaline (an arginase inhibitor), and cycloheximide (an inhibitor of protein translation)) enhanced the activity of mTORC1 (*Figure 1—figure supplement 2D,E,J*), suggesting that arginine acts as an intact molecule. Indeed, we observed that upon arginine starvation and recovery, arginine uptake is rapid and intracellular concentrations of arginine recover to steady state levels within 15 min of arginine re-addition. Within this same time frame, there were no changes in metabolites associated with arginine metabolism such as ornithine, citrulline, arginosuccinate, or fumarate indicating that arginine remains intact during this period (*Figure 1—figure supplement 2A,B*). Furthermore, following the addition of stable radiolabeled arginine (13C6,15N4) for 2 hr, we did not observe its incorporation into any other metabolites (*Figure 1—figure supplement 2C*), suggesting that arginine turnover is slow, at least in HeLa cells. These data suggest that free arginine is an important signal regulating mTORC1 activity.

## Arginine activates mTORC1 via TSC-Rheb axis

Several observations argue for a mechanism of arginine action different to that of leucine. First, deprivation of arginine but not leucine or isoleucine, significantly perturbed the growth factor-dependent input into mTORC1. This is evident from suppressed phosphorylation of ribosomal protein S6 kinase 1 (S6K1), eukaryotic translation initiation factor 4E-binding protein 1 (4E-BP1) and ULK1, with a concomitant increase in autophagy (*Figure 1A–C* and *Figure 1—figure supplement 3A–D*). Deprivation of arginine in the presence of growth factors limits mTORC1 activation to a level similar to that resulting from serum starvation. Addition of growth factors to leucine or isoleucine-deprived cells, however, permits maximal mTORC1 activation (*Figure 1A–C* and *Figure 1—figure supplement 3A–D*). Amino acids, particularly leucine, are known to signal to mTORC1 via the V-ATPase/Ragulator/Rag GTPase protein complexes and have been comprehensively demonstrated to control mTORC1 localization to the lysosome and its activity (*Carroll et al., 2015*). We observed that arginine did not affect mTOR localization in any cell line tested. While complete amino acid starvation and, to a lesser extent, leucine starvation, caused redistribution of mTORC1 to the cytoplasm, arginine deprivation led to a strong suppression of mTORC1 without a significant reduction in co-localization of mTOR with the lysosomal marker Lamp1 in HeLa, MEFs, and HEK293T cells (*Figure 1D*). Furthermore, overexpression of constitutively active Rag heterodimer (*Sancak et al., 2008*) did not completely rescue the effect of arginine starvation in the absence or presence of growth factors, thus suggesting that arginine may affect mTORC1 both via Rag-dependent and Rag-independent mechanisms (*Figure 1—figure supplement 3E,F*). Finally, the knockdown of membrane transporter, SLC38A9 that has recently been implicated in Rag GTPase-dependent lysosomal recruitment and activation of mTORC1 (*Wang et al., 2015*; *Rebsamen et al., 2015*; *Jung et al., 2015*) did partially perturb the recovery of mTORC1 following arginine starvation; however, there was no effect on the response of mTORC1 to arginine starvation either in the absence or presence of growth factors (*Figure 1—figure supplement 3G*).

Together, these data suggest that, in addition to an effect of arginine on mTORC1 via Rag GTPases/SLC38A9, this amino acid also plays an important role in regulating mTORC1 via the growth factor-regulated TSC-Rheb signaling axis. Upstream signaling events from growth factors via PI3K-Akt to TSC are not affected by deprivation of arginine (or complete amino acid or leucine), as assessed by Akt phosphorylation at both threonine 308 and serine 473 and downstream phosphorylation of TSC2 at serine 939 (*Figure 1—figure supplement 3H–K*). Similarly, arginine starvation is unlikely to suppress mTORC1 via activation AMPK, as neither in HeLa cells, where this pathway has low activity (*Corradetti et al., 2004*), nor in MEFs did we observe any increase in AMPK phosphorylation in response to arginine, leucine or complete amino acid withdrawal (*Figure 1—figure supplement 3L,M*). The loss of *TSC2*, however, did render MEFs insensitive to arginine (*Figure 1E* and *Figure 1—figure supplement 3N,O*), suggesting that arginine contributes to mTORC1 activity at the level of TSC. At the same time, sensitivity to leucine and complete amino acid starvation was preserved in *TSC2*[-/-] MEFs (*Figure 1E* and *Figure 1—figure supplement 3N,O*) in agreement with

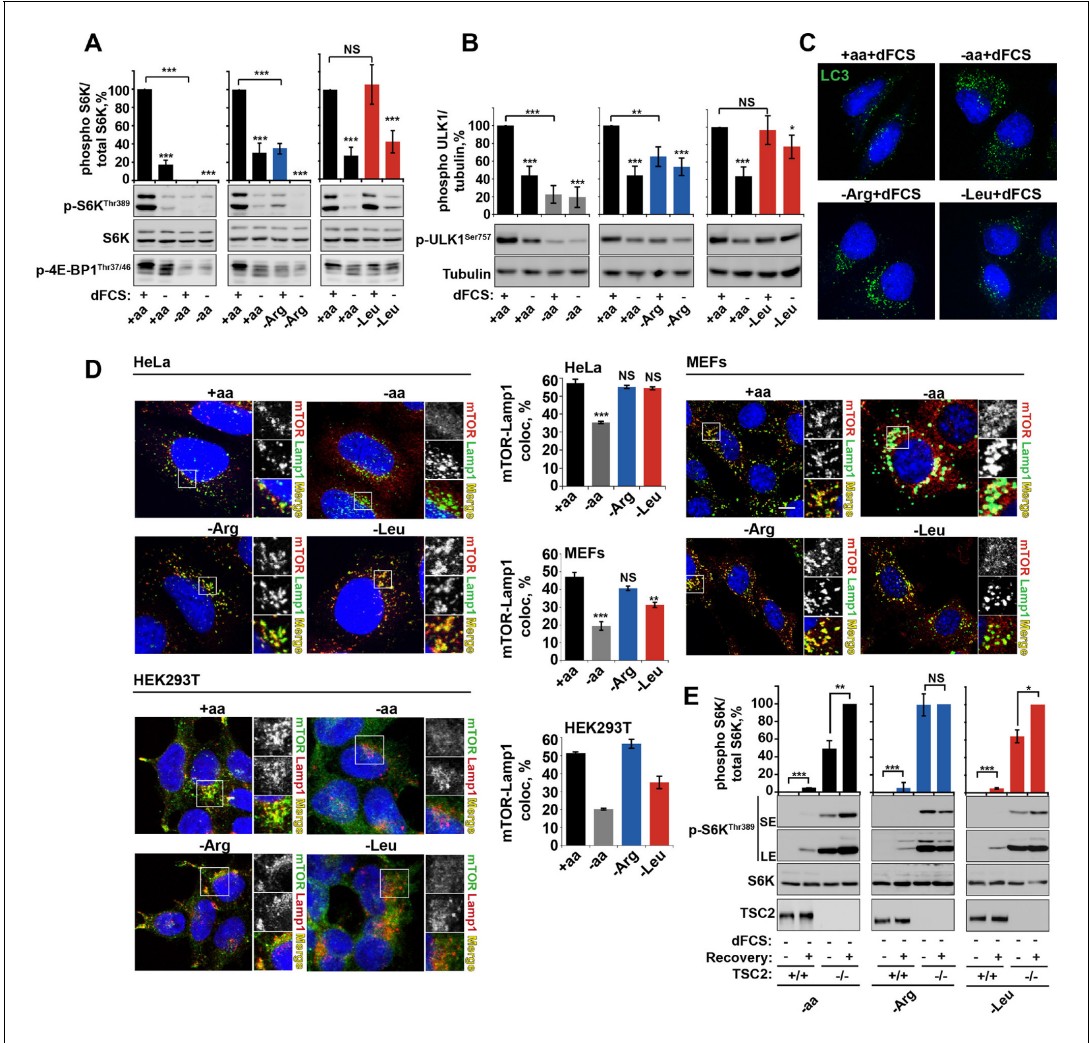

**Figure 1.** Arginine is a permissive factor for growth factor-dependent mTORC1 activity and acts via TSC2-Rheb axis. (A) Deprivation of arginine perturbs growth factor-mediated activation of mTORC1. HeLa cells were incubated with different amino acid mixtures in the presence/absence of dialyzed fetal calf serum (dFCS) as indicated. Lysates were subjected to immunoblotting for phosphorylation of S6K or 4E-BP1. (B, C) Arginine starvation activates autophagy. HeLa cells were treated as in A (for C also in the presence of chloroquine for 4 hr to block degradation of autophagosomes). Cells were immunoblotted for phosphorylation of ULK1 (B) or fixed and immunostained with antibodies against LC3 to detect autophagosomes (C). Scale bars: 10 μm. Nuclei are visualized in images with TO-PRO-3 iodide. (D) Arginine deprivation inhibits mTORC1 without causing its redistribution from lysosomal membranes to the cytoplasm. HeLa cells, MEFs, and HEK293T cells were incubated in the presence of amino acid mixtures as indicated. Cells were immunostained with antibodies against mTOR and Lysosomal-associated membrane protein 1 (Lamp1). Percentage co-localization between Lamp1 and mTOR was determined. (E) Loss of TSC2 renders mTORC1 insensitive to arginine deprivation. *TSC2*$^{+/+}$ and *TSC2*$^{-/-}$ MEFs were subjected to amino acid, arginine or leucine starvation. Where indicated, recovery was carried out by the re-addition of amino acids, arginine or leucine. Lysates were assessed for phosphorylation of S6K by immunoblotting. SE: short exposure; LE: long exposure. All graphs represent an average of at least three independent experiments (except mTOR localization in HEK293T cells which was carried out twice) and, where necessary, normalized to control treatment. Error bars represent s.e.m. *$p < 0.05$, **$p < 0.01$, ***$p < 0.005$. NS, not significant; Arg, arginine; Leu, leucine; aa, amino acids (complete set); dFCS, dialyzed FCS; MEF: mouse embryonic fibroblast.

The online version of this article includes the following figure supplement(s) for figure 1:

**Figure supplement 1.** Arginine and leucine are important mediators of mTORC1 activity in a wide range of cells.

**Figure supplement 2.** The metabolism of arginine does not contribute to the activation of mTORC1.

**Figure supplement 3.** Arginine contributes to mTORC1 activity via TSC2/Rheb signaling axis.

the idea that they are activating mTORC1 via other mechanisms such as Rag-dependent mTOR localization (*Figure 1D*).

Dynamic changes in TSC localization to lysosomes have been shown to control mTORC1 activity; however, it remains controversial whether it is regulated by growth factors or amino acids

(*Demetriades et al., 2014*; *Menon et al., 2014*). Therefore, we next investigated whether arginine could be involved in this process. In normal growth conditions, TSC2 was found in a diffuse pattern within both HeLa and MEF cells (*Figure 2A–D*). While serum starvation or arginine deprivation in the presence of dialyzed serum moderately promoted TSC2 localization to the lysosomal compartment, simultaneous starvation of arginine and growth factors resulted in an additive effect leading to very strong recruitment of TSC2 to Rab7- and Lamp1-positive late endosomal and lysosomal structures (*Figures 2A–D*, *Figure 2—figure supplement 1A,B*), which also correlates with a robust impairment of mTORC1 activity (*Figures 1A*). Recruitment of TSC2 to lysosomes was confirmed to be specific and no significant localization of TSC2 to other cellular organelles such as mitochondria, Golgi, or peroxisomes was observed (*Figure 2—figure supplement 1C–F*; [*Zhang et al., 2013*]). The effect of arginine is similar to that of all amino acids, while leucine did not influence lysosomal localization of TSC2 (*Figure 2—figure supplement 2A–D*). Replenishment of amino acids or arginine following starvation caused re-distribution of TSC2 to the cytoplasm (*Figure 2—figure supplement 2E*). Furthermore, the addition of arginine but not leucine to cells starved of all amino acids was sufficient to cause significant re-distribution of TSC2 to the cytoplasm (*Figure 2—figure supplement 2F*), demonstrating that arginine acts as a specific inhibitor of TSC2 recruitment to lysosomes. We further confirmed these immunofluorescence observations by lysosomal fractionation (enrichment of lysosomes was determined by western blot for multiple membrane and soluble proteins [*Figure 2—figure supplement 2G–H*]). Indeed, TSC2 was found to strongly accumulate upon a combination of serum and arginine (or all amino acid) starvation and, to a lesser extent, upon removal of a single stimulus (*Figure 2E*). The amount of Rheb in lysosomal fractions was not significantly affected by our starvation protocols (*Figure 2—figure supplement 2G*), suggesting that changes in localization of TSC in response to arginine may regulate Rheb function on lysosomes.

## Arginine suppresses recruitment of TSC to GTP-loaded Rheb

We next investigated whether arginine contributes to the regulation of TSC localization via either the Rheb or Rag GTPase-dependent mechanisms that have been described previously (*Menon et al., 2014*; *Demetriades et al., 2014*). In our system, we confirmed the central importance of Rheb in this process (*Menon et al., 2014*) by demonstrating that Rheb knock-down or inhibition of Rheb farnesylation (and therefore its membrane recruitment) completely prevented the arginine and growth factor starvation-induced re-localization of TSC2 (*Figures 3A–F* and *Figure 3—figure supplement 1A*). However, as reported before (*Demetriades et al., 2014*), we also found that expression of the dominant negative Rag constructs partially reduced lysosomal localization of TSC2 (*Figure 3—figure supplement 1B,C*). Consistent with its role in regulating mTORC1 via Rag GTPases, knock-down of SLC38A9 had a mild effect on TSC2 re-localization (*Figure 3—figure supplement 1D,E*). Furthermore, we tested whether mTOR itself is required for TSC recruitment to lysosomes. Knock-down of mTOR suppressed recruitment of TSC2 to lysosomes (*Figure 3—figure supplement 1F–H*). At the same time, rapamycin did not affect TSC2 recruitment suggesting that the activity of mTORC1 is not required for this translocation event (*Figure 3—figure supplement 1I*). As knockdown or hyperactivation of Rheb did not affect either total mTOR protein levels or lysosomal localization of mTOR (*Figure 3—figure supplement 2A–D*, also see [*Sancak et al., 2010*]), Rheb-dependent recruitment of TSC2 is unlikely to be indirectly mediated by perturbed mTOR localization. Taken together, these data strongly suggest that interaction with Rheb is the main mechanism of TSC2 recruitment to lysosomes but, in addition, the integrity of a larger lysosomal protein complex, including mTORC1 and Rag GTPases (*Demetriades et al., 2014*), is also important, possibly indirectly by regulating Rheb activity, localization or availability.

Small GTPase function is regulated by the GTP-nucleotide binding status of the protein, and indeed GTP-bound Rheb promotes mTORC1 kinase activity (*Long et al., 2005a*). By exploiting Rheb mutants that are either constitutively GTP-bound (R15G and N153T) or nucleotide-binding deficient (D60K and N119I) (*Li et al., 2004*; *Land and Tee, 2007*; *Urano et al., 2007* and *Figure 4—figure supplement 1A,B*), we went on to investigate whether the status of Rheb nucleotide loading is important for its interaction with TSC2 on lysosomal membranes. A model of the TSC2-Rheb complex indicates that mutations used here are unlikely to affect the Rheb-TSC2 interface other than by modifying nucleotide loading (*Figure 4—figure supplement 1A,B*). Interestingly, only Rheb·GTP (and wild-type although this is also predominantly GTP-bound) promoted TSC2 localization to lysosomes, and this was observed regardless of cellular nutrient status (*Figures 4A* and *Figure 4—figure*

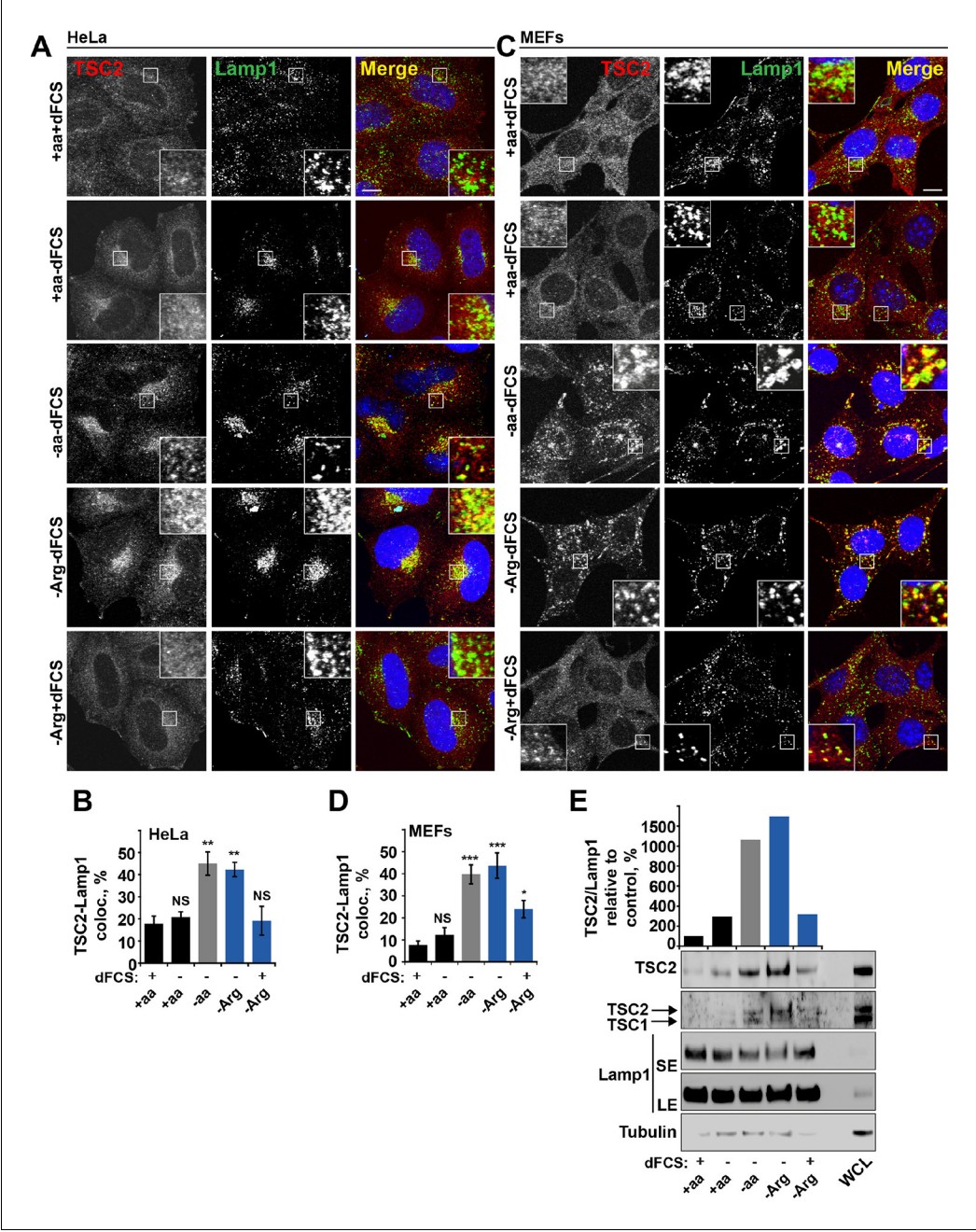

**Figure 2.** Arginine regulates TSC2 localization to the lysosome. (A–E) Starvation of arginine and growth factors cooperate to promote recruitment of TSC2 to lysosomes. HeLa cells (A, B) and MEFs (C, D) were subjected to different starvation protocols; full nutrient medium (+aa+dFCS), serum starvation (+aa-dFCS), arginine starvation (-Arg+dFCS), or starvation of serum and arginine (-Arg-dFCS). Cells were immunostained with antibodies against TSC2 and Lamp1 (A, C) and percentage co-localization between Lamp1 and TSC2 was determined (B, D). Scale bars: 10 μm. Nuclei are visualized in images with TO-PRO-3 iodide. (E) HeLa cells were subjected to different starvation protocols as indicated for 1 hr prior to harvesting. Lysosomes were isolated and equal amounts of protein in whole cell lysates (WCL) and lysosomal fractions were analyzed by immunoblotting for TSC2, TSC1, Lamp1 and Tubulin. SE: short exposure; LE: long exposure. All graphs represent an average of at least three independent experiments and error bars represent s.e.m. *p<0.05, **p<0.01, ***p<0.005.

The online version of this article includes the following figure supplement(s) for figure 2:

**Figure supplement 1.** TSC2 specifically localizes to late endosomal/lysosomal compartments during starvation.

**Figure supplement 2.** Arginine specifically regulates TSC2 localization to the lysosome.

*supplement 1C–E*). Lysosomal recruitment of TSC2 by Rheb·GTP in the presence of nutrients is likely

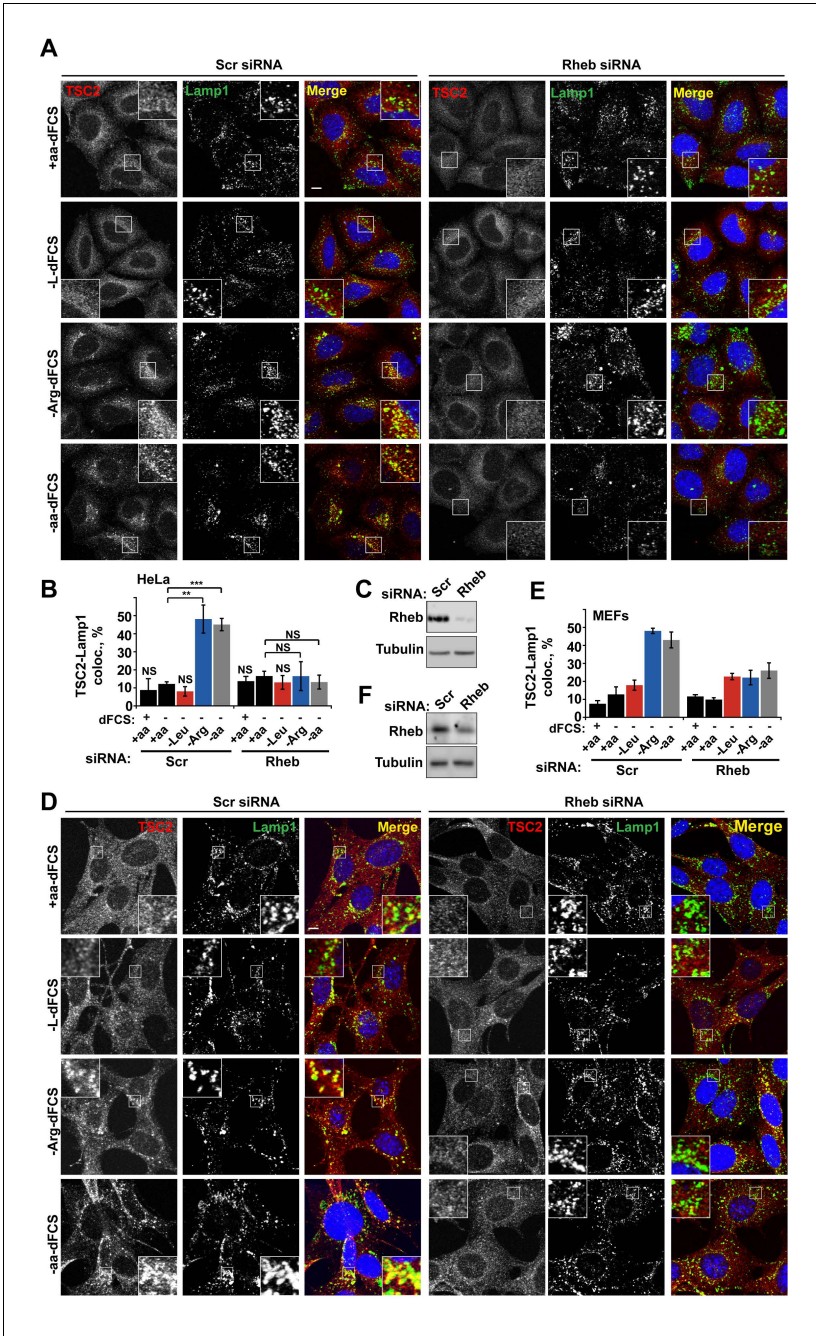

**Figure 3.** Rheb is required for the amino acid-dependent recruitment of TSC2 to lysosomal membranes. HeLa cells (**A–C**) and MEFs (**D–F**) were treated with scramble (Scr) or Rheb siRNA for 96 hr and subjected to different starvation protocols as indicated. Cells were fixed and immunostained with antibodies against TSC2 and Lamp1 (**A, D**). The percentage co-localization between Lamp1 and TSC2 was determined (**B, E**) and Rheb protein knock-down was assessed by immunoblot (**C, F**). Scale bars: 10 μm. Nuclei are visualized in merge images with TO-PRO-3 iodide. Graph represents an average of at least three independent experiments (**B**) or two independent experiments (**E**) and error bars represent s.e.m and s.d., respectively. *p<0.05, **p<0.01, ***p<0.005.
The online version of this article includes the following figure supplement(s) for figure 3:

**Figure supplement 1.** Rheb localization at lysosomes participates in TSC2 recruitment to lysosomal membranes during amino acid starvation.

**Figure supplement 2.** Rheb knock-down or hyperactivation does not affect mTOR localization.

to take place because the local concentration of amino acids is insufficient to interfere with overexpressed Rheb, consistent with previous studies of TSC2-Rheb·GTP interaction (*Castro et al., 2003*).

Combined with previous data (*Figures 3* and *Figure 3—figure supplement 1A*), these observations allowed us to conclude that Rheb is both necessary and sufficient (when GTP-bound) for the recruitment of TSC2 to lysosomes and arginine interferes with process. It was not clear, however, whether this interference was occurring at the surface of the lysosome. To investigate this, TSC2 was constitutively targeted to the cytoplasmic surface of lysosomes by a p18-derived lysosomal targeting signal (*Menon et al., 2014*) (*Figure 4B*). Interestingly, inactivation of mTORC1 signaling by either wild-type or constitutively lysosomal TSC2 was found to be suppressed by arginine (*Figure 4C*). This suggests that arginine interferes with Rheb-TSC2 interaction irrespective of lysosomal recruitment. To test this, we performed pull-down assays where immunoprecipitated Flag-Rheb was incubated with lysates of cells expressing V5-TSC2 and subjected to serum and amino acid starvation protocols (*Figure 4D*). In agreement with inhibition of Rheb-TSC2 interaction by arginine, deprivation of arginine but not leucine promoted strong binding of TSC2 to Rheb (*Figure 4D*). Moreover, an interaction of TSC2 (immunoprecipitated from mammalian cells) with recombinant GST-Rheb in vitro was perturbed by the addition of physiological concentrations of arginine (*Figure 4E*). These findings do not preclude an indirect effect of arginine via a putative sensor co-immunoprecipitated with TSC2; however, it strongly supports the role of arginine in regulating Rheb-TSC2 interaction. In agreement with binding assays, arginine also reduced the ability of TSC2 to promote hydrolysis of Rheb-bound GTP (*Figure 4F,G*).

## TSC inhibits GTP-loaded Rheb by a GAP-independent mechanism

Interestingly, despite the strong interaction of Rheb with TSC2 in the absence of arginine, a large fraction of Rheb (~40%) remains in GTP-bound active state (*Figure 4D–G*). This is consistent with data in vivo suggesting that a large fraction of Rheb remains in GTP-bound form even when mTORC1 is inactivated by starvation (*Long et al., 2005b*). Therefore, we hypothesized that a stable interaction between GTP-loaded Rheb and TSC2 may lead to steric hindrance or a conformational change thus preventing activation of mTORC1 by Rheb. In agreement with this idea, the increased binding of TSC to Rheb is at the expense of the interaction between Rheb and mTOR (*Figure 4—figure supplement 1F,G*), which would render mTORC1 inactive and explain severe inhibition of its activity seen in complete starvation conditions.

If TSC2 can inactivate Rheb by an additional, GAP-independent mechanism then TSC2 should still be able to inhibit mTORC1 activity driven by constitutively GTP-loaded Rheb mutants. Indeed, overexpression of TSC2 efficiently inhibited activation of mTORC1 by both wild-type and active Rheb mutants (R15G and N153T) both in the presence and absence of growth factors, despite resistance of these constructs to GAP activity of TSC2 (*Figures 4H* and *Figure 4—figure supplement 1A,B,H* and [*Urano et al., 2007*; *Li et al., 2004*; *Urano et al., 2005*; *Marshall et al., 2009*]). Similarly, if binding of TSC2 to Rheb is sufficient to inhibit its activity towards mTORC1, then GAP activity of TSC2 may be dispensable. This is indeed the case as overexpression of GAP-domain-dead TSC2 mutants (*Zhang et al., 2013*) efficiently suppressed activity of endogenous or overexpressed Rheb, and perturbed Rheb-mTOR binding, albeit to a lesser extent when compared to the wild-type construct (*Figure 4I* and *Figure 4—figure supplement 1I,J*).

Therefore, we propose that one of the mechanisms by which serum and amino acid (particularly arginine) starvation suppresses activity of mTORC1 is by increased TSC2-Rheb interaction on lysosomes, which results in a partial hydrolysis of GTP as well as in the shielding of remaining GTP-loaded Rheb from mTORC1.

## Arginine is a primary amino acid input into mTORC1

Finally, we tested the relative contribution of arginine-dependent mTORC1 regulation as compared to branched chain amino acids, sensing of which is believed to be the main function of mTORC1, in human embryonic stem cells (hESCs) and adult human cells differentiated from hESCs (*Figure 5—figure supplement 1A–E*). Interestingly, in hESCs mTORC1 activity was not suppressed by deprivation of leucine or isoleucine but was sensitive to arginine. This sensitivity to arginine remained an important limiting factor for mTORC1 activation in hESC-derived cells, including fibroblasts, neural precursors, neurons, and hepatocytes (*Figure 5A–E*). At the same time, sensitivity to leucine was

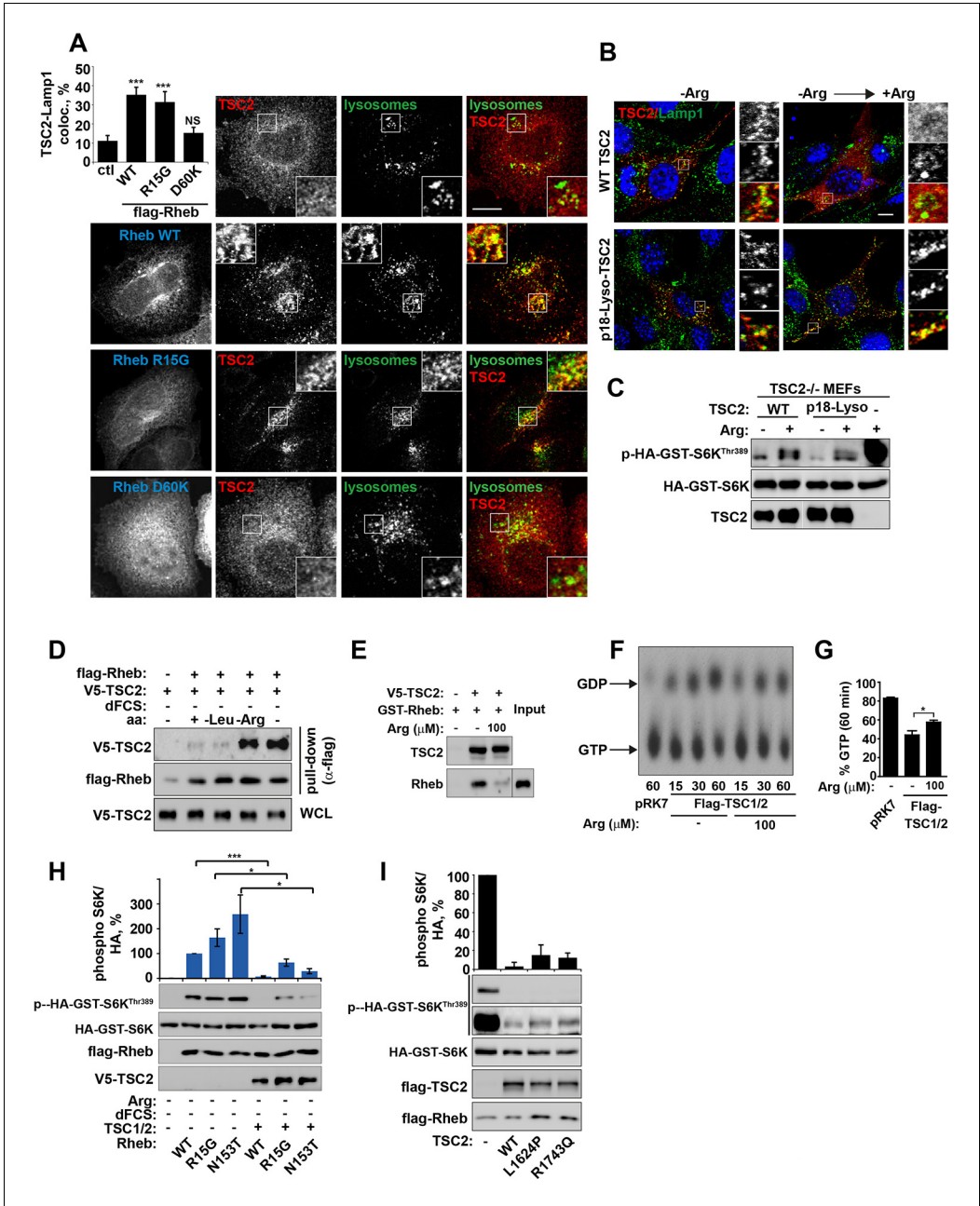

**Figure 4.** Arginine regulates the ability of TSC2 to bind and inhibit GTP-loaded Rheb independent of its ability to promote GTPase activity. (**A**) TSC2 recruitment to lysosomes is regulated by nucleotide-loading state of Rheb. HeLa cells were transfected with wild-type (WT), active Rheb (R15G), or inactive Rheb (D60K) overnight in the presence of GFP-lysosome BacMam. Cells were maintained in full nutrient medium, fixed and immunostained with antibodies against TSC2 and Flag-tag. The percentage of cells with lysosomal TSC2 was quantified. (**B, C**) Arginine suppresses activity of both wild-type and constitutively targeted to lysosomal membrane TSC2. *TSC2*$^{-/-}$ MEFs were transfected with either WT or p18-TSC2 (and HA-S6K (**C**)) overnight. Cells were deprived and replenished with arginine and either fixed and immunostained for Lamp1 and TSC2 (**B**) or subjected to immunoblot analysis of S6K phosphorylation (**C**). (**D**) Amino acids, specifically arginine regulate the interaction between Rheb and TSC2. HeLa cells were transfected with wild-type Flag-Rheb overnight, lysed and loaded onto anti-Flag beads. HeLa cells transfected with V5-TSC2 were subject to starvation protocols as indicated prior to lysis. Lysates were incubated with Rheb-loaded beads at 4°C for 1 hr. Samples were analyzed by immunoblotting to detect interaction of TSC2 with Rheb. (**E**) HEK293 cells were transfected with equal amounts of TSC1 and V5-tagged TSC2. TSC complexes were isolated by immunoprecipation with V5 agarose. Immunoprecipitates were incubated with recombinant GTP-loaded GST-Rheb in the presence or absence of 100 μM arginine. TSC2-Rheb binding was analyzed by immunoblot. (**F, G**) TSC complexes and Rheb were isolated from HEK293 lysates. GTP-loaded Rheb was incubated with TSC complexes in the presence or absence of 100 μM arginine. GTP/GDP loading of Rheb was assessed as described in the methods. (**H**) The TSC complex inhibits Rheb-mediated mTORC1 activity independent of GAP activity. Cells were transfected with Flag-wild-type Rheb, Flag-Rheb R15G or Flag-Rheb N153T with V5-TSC2 and HA-GST-tagged S6K overnight. Cells were subjected to arginine and serum starvation prior to

*Figure 4 continued on next page*

*Figure 4 continued*

lysis and immunoblotted for phosphorylation of S6K. (I) HeLa cells were transfected overnight with wild-type or GAP-deficient TSC2, Flag-Rheb and HA-GST-tagged S6K. Cells were lysed and immunoblotted for phosphorylation of S6K. Scale bars: 10 µm. All graphs represent an average of at least three independent experiments and error bars represent s.e.m., except for (E), which represents two repeats and error bars represent s.d. *p<0.05, **p<0.01, ***p<0.005.

The online version of this article includes the following figure supplement(s) for figure 4:

**Figure supplement 1.** Arginine contributes to the regulation of Rheb activity by TSC2.

acquired only after differentiation, although its depletion had lesser impact on mTORC1 in neurons and hepatocytes compared to that of arginine (*Figure 5A–E*). Together, these data suggest that arginine is the primary signal for the mTORC1 pathway in hESC and hESC-derived neurons and hepatocytes, while leucine-dependent regulation is an additional layer of control established only following differentiation.

## Discussion

Leucine and arginine represent the major contributors to amino acid-dependent mTORC1 activity. While the role of leucine, together with glutamine, in Rag-dependent activation of mTORC1 has been the subject of intense investigations, the role of arginine in mTORC1 activation has remained relatively understudied. Arginine can contribute to mTORC1 activity via the canonical amino acid sensing, Ragulator/Rag signaling pathway via the recently characterized low-affinity amino acid transporter SLC38A9 (*Wang et al., 2015*; *Rebsamen et al., 2015*; *Jung et al., 2015*). Our data suggest that arginine also works via the parallel, TSC2-Rheb signaling axis upstream of mTORC1 and stimulation of both inputs is required for the maximal activation of this signaling pathway (*Figure 6*). In particular, we show that arginine cooperates with growth factors and acts as a permissive factor for their activation of mTORC1. Growth factors and arginine interfere with the TSC2-Rheb interaction; while Akt-mediated growth factor signaling phosphorylates TSC2, arginine does not affect phosphorylation of Akt, at least on the sites tested in this study. Instead, arginine appears to act as an intact molecule at a site more proximal to the TSC2-Rheb interaction, whether acting directly or via a putative sensor molecule.

Interestingly, removal of both serum and arginine is required for complete inhibition of mTORC1 signaling, suggesting that either stimulus can support low levels of basal mTORC1 activity (*Figure 1A*). Our measurements of TSC2 localization and nucleotide loading of Rheb help to explain this mechanistically. We propose that in response to serum or arginine starvation TSC2 transiently interacts with Rheb in a GTP-bound conformation on the surface of lysosomes promoting its GTPase activity. However, due to high basal GTP levels of Rheb (*Li et al., 2004*), the GAP activity of TSC2 is not sufficient for complete GTP hydrolysis to GDP leaving a substantial fraction of Rheb in a GTP-bound state. This residual-active GTP-bound Rheb allows for the low level of mTORC1 activation in conditions of serum or arginine starvation. On the other hand, deprivation of both stimuli results in a complete suppression of mTORC1 activity, which is associated with a stable interaction of TSC with Rheb on lysosomes. This complex formation serves to sequester Rheb and to prevent its interaction with mTORC1 and its activation. By regulating Rheb-TSC2 interaction, arginine acts as a switch for upstream signaling inputs from growth factors.

Previous reports describing the spatial regulation of TSC2 as a mechanism controlling mTORC1 activity have demonstrated that various inputs such as mechanical stimulation, growth factors and amino acids can contribute to the tight regulation of the lysosomal localization of TSC2 (*Cai et al., 2006*; *Jacobs et al., 2013*; *Demetriades et al., 2014*; *Menon et al., 2014*). *Demetriades et al. (2014)* argue the importance of an amino acid/Rag GTPase-dependent recruitment of TSC2 to lysosomes, while instead *Menon et al. (2014)* demonstrate that deprivation of growth factors, the classical regulators of TSC activity, controls its recruitment to lysosomes. Our findings begin to reconcile these differences by identifying that amino acids (specifically arginine) cooperate with growth factors to fine tune the spatial and temporal regulation of TSC and therefore mTORC1. We show that both arginine and growth factors suppress strong recruitment of TSC2 to lysosomes by interfering with stable TSC2-Rheb interaction as described by *Menon et al. (2014)*. In agreement with *Demetriades et al. (2014)*, we also find a role of Rag GTPases in this process (*Demetriades et al.,*

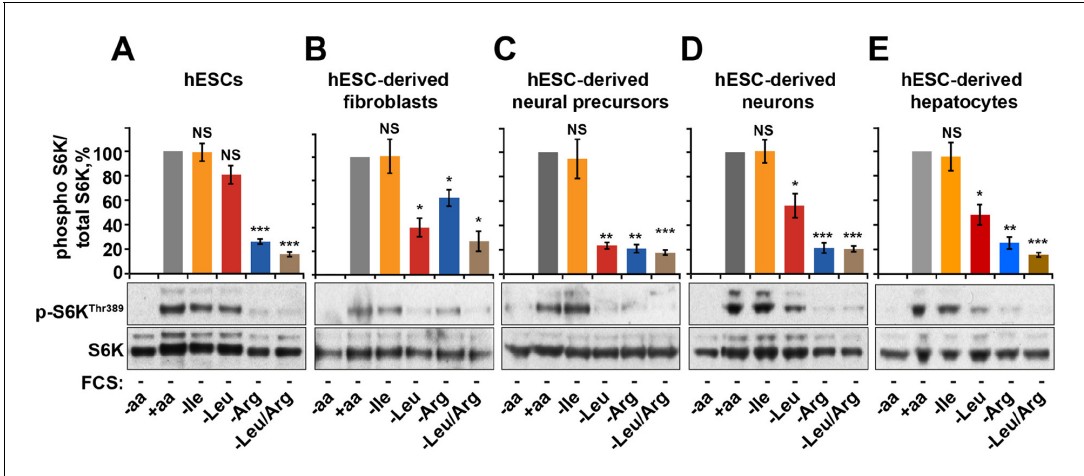

**Figure 5.** Arginine, but not leucine is essential for mTORC1 activity in embryonic stem cells. Human embryonic stem cells (hESCs) were maintained (**A**) or differentiated to fibroblasts (**B**), neural precursors (**C**), neurons (**D**), or hepatocytes (**E**). hESCs and the derived lineages were starved of serum and indicated amino acids (isoleucine (Ile), leucine (Leu), arginine (Arg), or both leucine and arginine (Leu/Arg)). Lysates were subjected to immunoblotting for phosphorylation of S6K. All graphs represent an average of at least three independent experiments and error bars represent s.e.m. *p<0.05, **p<0.01, ***p<0.005.

The online version of this article includes the following figure supplement(s) for figure 5:

**Figure supplement 1.** Embryonic stem cell differentiation.

2014), which are likely to be acting via mTORC1, thus suggesting that the recruitment of TSC to lysosomes is a complex process mediated not only by Rheb but also by its interacting partners. At the same time, by proposing a dual mechanism of Rheb inhibition by TSC (GTPase activation and sequestration into a stable complex which prevents the interaction of Rheb with mTOR as discussed above, *Figure 4*), our interpretation of TSC2 localization phenotypes conceptually extends the conclusions of others.

The ability of arginine to regulate mTORC1 activity is fundamentally important, and it is the only amino acid essential to both undifferentiated stem cells and differentiated lineages (*Figure 5*). The fact that mTORC1 sensitivity to leucine is acquired only after differentiation might reflect the need to integrate an increased complexity of extracellular stimuli to mTORC1. Arginine is an essential amino acid during embryogenesis which is a period of active growth and proliferation largely driven by mTORC1 (*Hentges et al., 2001*; *Wu et al., 2009*; *Wu et al., 2013*), and we speculate that this underlies the dependence of mTORC1 on this amino acid. Growth-promoting signals in the absence of arginine would lead to defective protein synthesis, which would explain the need for coordination between growth factor and arginine availability. We acknowledge, however, that other, arginine-independent molecular mechanisms contribute to mTORC1 regulation, and, together, these serve to provide extremely tight spatial and temporal regulation of mTORC1 signaling in response to nutrients. Indeed, while arginine, leucine, and glutamine are sufficient for mTORC1 signaling, the level of activation achieved by these amino acids is very low compared to an entire set of free amino acids. Understanding the mechanisms by which mTORC1 is sensing the entire complement of amino acids provides an exciting challenge for future investigations.

## Materials and methods

All chemicals were from Sigma Aldrich (St. Louis, MO) unless indicated otherwise.

### Cell lines

HeLa cells, HEK293T, the osteosarcoma U2OS, primary human fibroblasts (MRC5), TSC2-deficient ($TSC2^{-/-}$) and wild-type ($TSC2^{+/+}$) mouse embryonic fibroblast cell lines (MEFs) (a kind gift from D. Kwiatkowski, Harvard University, Boston) and neuroblastoma SK-N-SY were cultured in Dulbecco's Modified Eagle Medium (DMEM; Sigma D6546) with 2 mM L-glutamine, 10% Foetal Bovine Serum

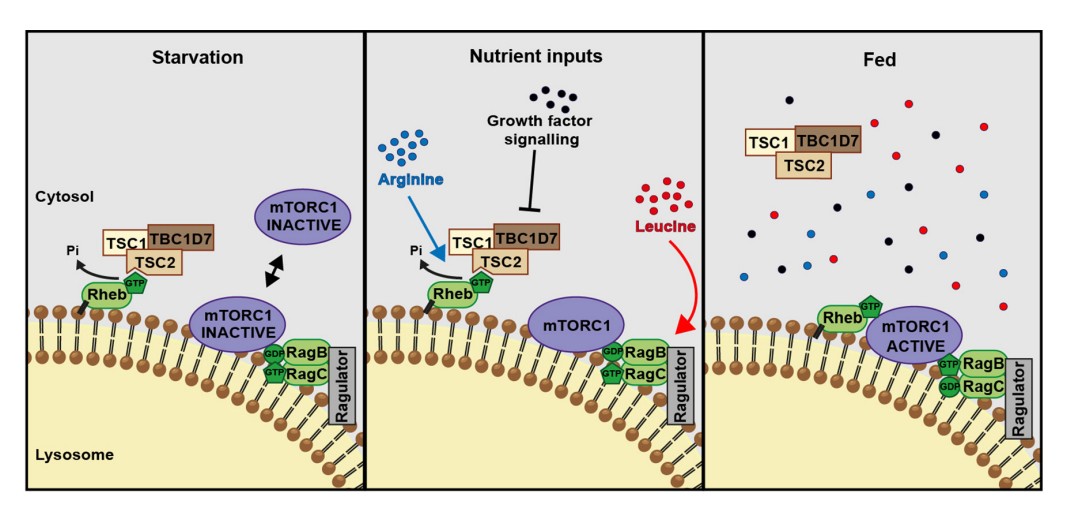

**Figure 6.** A model of amino acid-dependent mTORC1 regulation. Amino acids, most notably leucine activates mTORC1 via Rag GTPases and recruitment of mTOR to lysosomes. Arginine cooperates with growth factors to prevent interaction between TSC and Rheb on lysosomes which allows interaction between Rheb and mTORC1. See text for further details.

(FBS), 100 Units/ml penicillin-streptomycin at 37°C, 5% $CO_2$. Primary mouse neurons were cultured in Neurobasal medium containing b27 supplement (2%), glutamine (0.5 mM), and pen/strep (1%).

## Human embryonic stem cell maintenance and differentiation

Human embryonic stem cells (hESCs) WIBR3 (Whitehead Institute Center for Human Stem Cell Research, Cambridge, MA) were cultured as described previously (*Lengner et al., 2010*). Briefly, hESCs were maintained on mitomycin C -inactivated MEF feeder layers in hESC medium [DMEM/F12 (Life Technologies, Carlsbad, CA) supplemented with 15% FBS (Hyclone, Logan, UT), 5% Knock-Out Serum Replacement (Life Technologies), 1 mM glutamine (Life Technologies), 100 U/ml penicillin/streptomycin (Life Technologies), 100 U/ml penicillin/streptomycin (Life Technologies), 1% non-essential amino acids (Life Technologies), 0.1 mM β-mercaptoethanol, 4 ng/ml FGF2 (R&D Systems)]. Cultures were passaged every 5 to 7 days either manually or enzymatically with 1 mg/ml collagenase type IV (Life Technologies). Differentiation into hepatic-like cells was induced as described previously (*Si-Tayeb et al., 2010*). Briefly, hESCs were plated on matrigel-coated plates as single cells and cultured in the presence of ROCK inhibitor for 24 hr. After reaching 80% confluency, differentiation was initiated following published protocols (*Si-Tayeb et al., 2010*). As described previously, fibroblasts and neural progenitors (NPs) were derived using an embryoid body (EB)-based protocol (*Xu et al., 2004*; *Marchetto et al., 2010*). NPs were differentiated into neurons as described previously (*Soldner et al., 2009*; *Chambers et al., 2009*).

## Transfection methodologies

Cells were transfected using FugeneHD (Promega, Madison, WI) or Lipofectamine 2000 (Life Technologies) according to the manufacturer's protocols for 24 hr prior to lysis.

## Plasmids

The following plasmids were purchased from Addgene (Cambridge, MA): pcDNA3-flag Rheb[N153T] (#19997) (*Urano et al., 2007*) and HA-GST p70S6K (#15511), pRK5-HA-GST RagB Q99L (# 19303), pRK5-HA-GST RagC S75L (# 19305). V5-TSC2 was a kind gift from M. Nellist (Erasmus MC University Medical Center, Rotterdam, The Netherlands) (*Hoogeveen-Westerveld et al., 2012*). Lyso-TSC2 fused to the lysosomal-targeting sequence of p18 was a kind gift from B. Manning (Harvard School of Public Health, Boston, MA) (*Menon et al., 2014*). Flag-tagged TSC2 constructs (R1743Q and L1624P) have been described previously (*Zhang et al., 2013*). Myc-mTOR (*Korolchuk et al., 2011*) and GFP-Rab7 (*Seaman, 2004*) have been described previously. pcDNA-flag-Rheb wild-type has been described previously (*Dunlop et al., 2011*). Wild-type Rheb, the active mutant R15G, and the

dominant negative mutant D60K were generated by mutagenesis of the threonine in pcDNA3-flag RhebN153T to asparagine followed by sequential mutation of arginine at residue 15 to glycine (R15G) or aspartic acid at residue 60 to lysine (D60K) using Stratagene Quickchange mutagenesis kit (Stratagene, San Diego, CA) as per company instructions. Primers: T153N to generate wild-type Rheb 5'-ccacagcagtctgattttctttagcagaagattccaaaaaag-3' and 5'-cttttttggaatcttctgctaaagaaaatca-gactgctgtgg-3'. Rheb R15G 5'- ttccccacagacccgtagcccaggatc-3' and 5'- gatcctgggctacgggtctgtggg-gaa-3'. Rheb D60K 5'-tggacaagaatatcatcttcaacttgtaaaaacagccgggcaagat-3' and 5'-atcttgcccggctgtttttacaagttgaagatgatattcttgtcca-3'.

## siRNA

ON-TARGETplus SMARTpool siRNA against human Rheb (#6009), mouse Rheb (#19744 Dharmacon, Lafayette, CO), human RARS (#5917), human SLC38A9 (#007337) and non-targeting SMARTpool siRNA(D-001810-04) were purchased from Dharmacon. Four individual oligos against human mTOR (Flexitube FRAP1_4–7, Qiagen, Germany) were pooled to a final stock concentration of 20 µM. Final siRNA concentrationsof 100 nM were used for 96 hr for silencing and transfections were carried out using Lipofectamine 2000 as per company instructions.

## Starvation and recovery protocols

Cells were grown in 12- or 6-well plates until 80% confluent. Serum starvation was carried out over-night, amino acid starvations were 1 hr and, where indicated, amino acid starvation in the presence of dialyzed FCS (dFCS) was carried out for 1 hr. For amino acid starvation, cells were washed briefly in PBS followed by 60 min incubation at 37° C with RPMI without amino acids (US Biologicals, Salem, MA) or in the presence of various amino acid mixtures (and, where stated, dFCS). Recovery experiments were carried out by the addition of amino acids for 30 min before lysis. 100x stocks of amino acid mixtures were prepared from individual powders and diluted to 1x in RPMI without amino acids to the final concentration as found in DMEM media. Dialyzed FCS was purchased as <1 kDa cut-off (Dundee Cell Products, UK) and further dialyzed with cassettes of <3 kDa cut-off. Where noted, cells were incubated with insulin at a final concentration of 10 µg/ml and EGF (Peprotech, Rocky Hill, NJ) at 100 ng/ml. As a positive control for activation of AMPK, cells were incubated in DMEM without glucose (Life Technologies; supplemented with 4 mM L-glutamine, 10 mM D-galactose, 10 mM HEPES, 1 mM sodium pyruvate, penicillin and streptomycin and 10% FBS) for 1 hr.

HeLa cells were incubated with 1 µM of the farnesyl-transferase inhibitor (FTI), lonafarnib (Caymen Chemicals, Ann Arbor, WI; #11746) or DMSO control for 24 hr. Cells were then incubated in the presence or absence of amino acids (in the presence of lonafarnib) before being fixed and immunos-tained. HeLa cells were incubated with rapamycin (100 nM) for 1 hr pre-treatment in full nutrient medium. Cells were then incubated in the presence or absence of amino acids in RPMI without amino acids for 1 hr. Cells were lysed for immunoblot to confirm inhibition of mTOR activity or fixed and stained for TSC2 and Lamp1.

## Immunoblotting

Cells were lysed in RIPA Buffer (50 mM Tris-HCl pH 7.4, 150 mM NaCl, 1% NP40, 0.5% sodium deox-ycholate, 0.1% SDS and supplemented with Halt protease and phosphatase inhibitors (Thermo Sci-entific, Waltham, MA; #1861280)) on ice. Lysates were centrifuged at 13,000 rpm for 10 min at 4°C and protein concentration was measured using DC protein assay (Bio-Rad, Hercules, CA; #500–0112). Equal amounts of protein (20–40 µg) were subjected to SDS-PAGE and immunoblotted, as previously described (*Korolchuk et al., 2011*) and (*Smith et al., 2005*). For immunoblotting of SLC38A9, samples were prepared as described in (*Rebsamen et al., 2015*), samples were prepared in sample buffer with reducing agent and incubated at room temperature for 15 min rather than boiled at 100°C. The following antibodies were used: rabbit anti-mTOR (#2972, 1:1000), rabbit anti-phospho S6K$^{Thr389}$ (#9205S, 1:1000), rabbit anti-S6K (#9202, 1:1000), rabbit anti-phospho S6$^{Ser235/236}$ (#4856, 1:2000), rabbit anti-S6 (#2217, 1:2000), rabbit anti-phospho ULK1$^{ser939}$ (#3615, 1:1000), rabbit anti-phospho Akt$^{Ser473}$ (#9271, 1:1000), rabbit anti-phospho Akt$^{Thr308}$ (#4056, 1:1000), rabbit anti-phospho 4E-BP1$^{Thr37/46}$ (#2855, 1:1000), rabbit anti-phospho AMPK$^{Thr172}$ (#2535, 1:1000), rabbit anti-Na+K+-ATPase (#3010. 1:1000), rabbit anti-TSC1 (#4906, 1:1000), rabbit anti-TSC2 (#4308, 1:1000) and rabbit anti-phospho TSC2$^{Ser939}$ (#3615S, 1:1000) were all purchased from Cell Signaling

Technologies (Danvers, MA). Other antibodies used in this study include mouse anti-alpha-tubulin (12G10, DSHB, Iowa City, IA), goat anti-Rheb (clone C-19 #sc-6341, Santa Cruz, Dallas, TX, 1:500), mouse anti-flag (clone M2, #F3165, Sigma, 1:1000), mouse anti-Lamp1 (clone H4A3, Abcam, Cambridge, MA, 1:1000), mouse anti-V5 (# R960-25, Life Technologies, 1:5000), mouse anti-myc (Roche, Switzerland, 1:1000), mouse anti-HA (Covance, Princeton, NJ, 1:1000), mouse anti-GM130 (#610822, BD Bioscience, UK, 1:1000), mouse anti-PMP70 (#SAB4200181, Sigma, 1:2000) and mouse anti-SLC38A9 (#HPA043785, Sigma). Secondary antibodies conjugated to horseradish peroxidase (HRP) were all used at 1:5000 for 1 hr at room temperature. Clarity western ECL substrate (Bio-Rad) was used to visualize chemiluminescence on LAS4000 (Fujifilm). Quantification of blots was carried out using ImageJ (NIH).

## Immunofluorescence

Where indicated, lysosomes were visualized with CellLight Lysosomes-GFP, BacMam 2.0 (Life Technologies) and mitochondria were labeled with MitoTracker green (Life Technologies) by incubation overnight. Immunofluorescence was carried out essentially as described previously (Korolchuk et al., 2011). Briefly, cells were fixed in 4% formaldehyde in PBS for 10 min at room temperature. Cells were permeabilized with 0.5% Triton X-100 (or methanol at −20°C for LC3 staining) for 5 min at room temperature. Following 1 hr of blocking in 5% normal goat serum/ PBS 0.05% Tween-20 (Tween-20 was omitted for LC3 staining), cells were incubated with primary antibodies overnight at 4°C. Primary antibodies used in this study include anti-mTOR (#2972 Cell Signaling Technologies, 1:200), anti-TSC2 (#4308 Cell Signaling Technologies, 1:1000), mouse anti-PMP70 (#SAB4200181, Sigma, 1:1000), mouse anti-GM130 (#610822, BD Bioscience, 1:500), mouse anti-LC3 (Enzo/Nanotools, Germany; #LC3-2G6, 1:250), mouse anti-Lamp1 (Abcam, 1:1000), rat anti-Lamp1 (#1D4B, DSHB, 1:1000) anti-flag (Sigma, 1:1000), anti-V5 (Life Technologies, 1:1000), anti-HA (Covance, 1:1000), Oct4 (#sc-5279, Santa Cruz, 1:500), Nanog (#AF1997, R&D Systems, Minneapolis, MN, 1:500), SSEA4 (#MC-813-70, DSHB, 1:20), Nestin (#AB5922, Millipore, Germany, 1:500), Pax6 (#PRB-278P, Covance, 1:250), Tuj1 (#MMS-435P, Covance, 1:1000), AFP (#A8452 Sigma-Aldrich, 1:1000), HNF4α (#sc-6556, Santa Cruz, 1:500). Cells were washed three times and incubated with the appropriate secondary antibodies for 1 hr at room temperature (Life Technologies, 1:1000). Cells were washed and nuclear DNA was stained by incubation with TO-PRO-3 iodide (Life Technologies, 1:3000) for 10 min at room temperature. Coverslips were mounted on slides with Prolong Gold antifade reagent (Life Technologies).

## Lysosome isolation

HeLa cells were seeded in 10-cm$^2$ plates until 80–90% confluent. Cells were treated with various starvation protocols as indicated before lysis. Lysosomes were isolated with the Lysosomal enrichment kit for tissue and cultured cells as per company instructions (Thermo Scientific). Briefly, cells were scraped into cold Buffer A as provided with the kit, supplemented with protease and phosphatase inhibitors on ice. Lysates were vortexed at maximum speed for 5 s and incubated on ice for 2 min. Cells were lysed by sonication (lysis efficiency was optimized by visual comparison with non-lysed control) and equal volume of Buffer B (provided by kit) was added. Intact cells were removed by centrifugation at 500 x *g* for 10 min at 4°C. OptiPrep gradients were prepared with reagents provided in the kit, at the concentrations indicated in the kit. Samples were prepared with OptiPrep Cell Separation Media to a final concentration of 15% and overlayed on the gradient. Samples were centrifuged at 145,000 x *g* for 2 hr at 4°C. The top layer was removed and diluted with PBS before being subject to another centrifugation step at 18,000 x *g* for 30 min at 4°C. The subsequent pellet was washed with PBS and centrifugation was repeated as in the previous step. The lysosome-enriched pellet was resuspended in RIPA buffer, subject to protein quantification and Western blot analysis.

## Pull-down assay

HeLa cells were seeded in 10 cm$^2$ plates 24 hr prior to transfection with either pRK5-flag Rheb or V5-TSC2 and/or myc-mTOR. First, Rheb-expressing cells were lysed (buffer 1 (mTOR-Rheb) 20mM Tris pH 8.0, 10 mM MgCl, 0.3% CHAPS and 2x Halt protease and phosphatase inhibitors or buffer 2 (TSC2-Rheb) 40 mM Tris pH 7.4, 10 mM MgCl, 5 mM EGTA, 25 mM NaCl, 0.2% CHAPs, 2x Halt protease and phosphatase inhibitors [Sancak et al., 2007; Smith et al., 2005; Long et al.,

2005a; Sun et al., 2008; Castro et al., 2003; Rebhun et al., 2000]) on ice and centrifuged at 13,000 rpm for 10 min. Lysates were incubated with pre-washed and equilibrated anti-flag M2 magnetic beads (Sigma Aldrich, #M8823) for 1 hr at 4°C with constant rotation. The Rheb-loaded beads were washed four times in lysis buffer. TSC2/mTOR transfected cells were incubated in the presence or absence of arginine for 1 hr prior to lysis. Lysates were centrifuged at 13,000 rpm for 10 min at 4°C and samples were collected (5%) for subsequent analysis of protein expression levels. Lysates were incubated with Rheb-loaded beads (20 µl slurry/assay) for 1 hr at 4°C with constant rotation. Beads were washed twice with lysis buffer and the pulled-down protein was eluted from the beads by incubation with 25 µl 0.2 M glycine-HCl pH 2.5 for 10 min at room temperature. Eluent was neutralized by the addition of 2.5 µl Tris-HCl pH 8.8. The samples were then mixed with sample buffer and boiled at 100°C for 5 min before being subjected to western blot analysis.

## Liquid chromatography-mass spectrometry

LC-MS was carried out as previously described (Labuschagne et al., 2014; Maddocks et al., 2013). Briefly, HeLa cells were seeded (in triplicate) in six-well plates and cultured in standard DMEM until 90% confluent. Cells were serum starved overnight and subjected to amino acid starvation protocols as indicated. Alternatively, cells pre-incubated with 10 mM L-norvaline, 100 µM, ADMA, 1 mM L-citrulline, control siRNA or RARs siRNA (see above for treatment conditions) were washed twice with PBS and incubated in the presence or absence of labeled arginine (13C6, 15N4, [#CNLM-539-H, CK gas]) for 2 hr. Cells were washed once with cold PBS and lysed (50% methanol/30% acetonitrile/ 20% dH$_2$0) at a concentration of $2 \times 10^6$ cells per ml. Samples were vortexed for 45 s and centrifuged at 13,000 rpm. LC-MS was carried out as described (Labuschagne et al., 2014; Maddocks et al., 2013).

## Rheb nucleotide binding

HeLa cells were grown on 10-cm$^2$ plates and were incubated in 5 ml of phosphate-free medium containing 0.5 mCi [$^{32}$P]orthophosphate for 3 hr prior to lysis. Flag-tagged Rheb was immunoprecipitated for 1 hr at 4°C with anti-Flag antibodies bound to protein G-Sepharose. Immunoprecipitates were washed twice each with both buffer A (50 mM HEPES (pH 7.4), 100 mM NaCl, 10 mM MgCl$_2$, 1 mg/ml BSA, 1 mM DTT, 1% Triton) and buffer B (50 mM HEPES [pH 7.4], 100 mM NaCl, 10 mM MgCl$_2$, 0.1% Triton) in the presence of protease inhibitors. [$^{32}$P]-radiolabeled GTP and GDP were eluted from Rheb using 20 µl Rheb elution buffer (0.5 mM GDP, 0.5 mM GTP, 5 mM DTT, 5 mM EDTA, 0.2% SDS) at 68°C for 20 min and then resolved by thin layer chromatography on polyethyleneimine cellulose with KH$_2$PO$_4$.

## Rheb GAP assays

The GTPase-activating protein assays were carried out as previously described (Tee et al., 2003).

## GST-Rheb 1–184 protein expression and purification

An N-terminal GST fusion protein of full length human Rheb (GST-Rheb) was expressed in Rosetta2 (DE3) E. coli (Novagen, Merck Millipore, Billerica, MA) using a pGEX-4T-2 vector (Tee et al., 2003), grown in 2xYT media and induced with 0.5 m IPTG at 37°C for 3 hr. Cells were resuspended in 20 mM Tris pH 8.0, 500 mM NaCl, 2 mM DTT with EDTA-free protease inhibitors (Pierce, Thermo Fisher Scientific) and lysed by sonication. Initial purification was achieved by glutathione Sepharose 4B (GE Healthcare, UK) affinity chromatography, with elution in 20 mM Tris pH 8.0, 200 mM NaCl, 2 mM DTT, 10 mM reduced glutathione. GST-Rheb was further purified by anion exchange chromatography (GE Healthcare), concentrated (Millipore Amicon Ultra-4), flash-frozen in liquid nitrogen and stored at −80°C in 20 mM Tris pH 8.0, 300 mM NaCl, 5 mM MgCl$_2$, 2 mM DTT at 4 mg/ml. Protein samples were analysed by SDS-PAGE using the Bolt Bis-Tris system with SimplyBlue SafeStain (Thermo Fisher). Protein concentrations were measured using a Nanodrop 1000 spectrophotometer (Thermo Scientific) with extinction coefficients and molecular weights determined by ExPASy ProtParam (Bairoch et al., 2005)

## Rheb-TSC2 in vitro binding in the presence of arginine

In this study, $5 \times 10^6$ HEK293T cells were transfected in 10 cm dishes overnight before 5 µg each of Flag-tagged TSC1 and V5-tagged TSC2 were transfected for a further 24 hr. Cells were harvested in 1 ml lysis buffer (20 mM Tris pH 7.4, 150 mM NaCl, 1 mM $MgCl_2$, 1% NP-40, 10% glycerol, 1 mM DTT and 2x protease/phosphatase inhibitors). Lysates were incubated on ice and centrifuged at 13,000 rpm for 10 min at 4°C. Samples were incubated with 200 µl V5-conjugated agarose (Sigma) for 2 hr at 4°C. Beads were washed 2x in pull-down buffer (20 mM Hepes pH 7.4, 150 mM NaCl, 1 mM EDTA, 5 mM $MgCl_2$, 0.5% NP-40, 10 mg/ml BSA, 1 mM DTT and 2x protease/phosphatase inhibitors) and once in Rheb-loading buffer (20 mM Hepes pH 8.0, 200 mM NaCl, 5 mM $MgCl_2$, 10 mM EDTA). Equal volume of beads was aliquoted into separate, experimental tubes. Recombinant GST-Rheb1-184 was defrosted quickly and centrifuged at 13,000 rpm for 20 min at 4°C. GST-Rheb was diluted in Rheb-loading buffer and loaded with non-hydrolysable GTP (GTPγS [Cytoskeleton, Denver, CO]) by the addition of 0.2 mM GTPγS and incubated at 30°C for 10 min, followed by the addition of 20 mM $MgCl_2$. Reactions were prepared by incubating 2 µg/ml GST-Rheb either with or without 100 µM arginine. The pH of the buffer was adjusted to 7.4 before the addition of equal amounts of TSC2 agarose beads. Reactions were incubated for 1 hr at 4°C with rotation and beads were washed three times in TSC2 wash buffer. Immunoprecipitates were eluted by incubation with 0.2 M glycine pH 3 for 10 min at room temperature. Sample buffer was added to the eluted samples, boiled and analysed by western blot.

## Modeling of Rheb-TSC2 and in silico mutagenesis and electrostatics calculations

The structure of the Rheb-TSC2 complex was predicted using the X-ray crystal structure of the human Rap1-Rap1GAP complex (PDB: 3BRW) (*Scrima et al., 2008*) as a template. It was assumed that, because of the conservation of mechanism in small G-Protein-GAP systems, the two components of the Rheb-TSC2 complex would be oriented relative to one another in the same way, so an interface of approximately correct structure would be produced by superposing each component on its homologue in the Rap1-Rap1GAP complex.

TSC2: The model structure of the TSC2 GAP domain was generated with the SWISS-MODEL server (*Arnold et al., 2006*) (http://swissmodel.expasy.org/). The stoichiometry of Rap1-Rap1GAP in 3BRW is 1:3, but on examination of the complex it is clear that it is the B Rap1GAP chain that is bound in the correct orientation to assist in catalysis. Accordingly, the B chain was used as the template. The server was used in alignment mode with a manually adjusted alignment; the alignment was based on that automatically generated, and in the C-terminus, that used to build a pre-existing TSC2 model in the SWISS-MODEL repository (*Kiefer et al., 2009*) which had used the unbound Rap1GAP (PDB:1SRQ) as a template. The sequence ID is 21%, although higher in the N-terminal half. The bound TSC2 model returned by the server contains almost the entire GAP domain (residues 1525–1756) and is already in the coordinate frame in which it is superposed on the template.

Rheb and complex: The GTP-bound structure of human Rheb (PDB:1XTS) (*Yu et al., 2005*) was used, and was superposed on human Rap1 (PDB:3BRW chain D) using the PDBeFold structure-based superposition server (*Krissinel and Henrick, 2004*) (http://www.ebi.ac.uk/msd-srv/ssm/cgi-bin/ssmserver). The two components were then concatenated. The resulting complex was refined by energy minimization with GROMACS (*Van Der Spoel et al., 2005*) to remove steric clashes in the interface (bound GTP was removed to do this and then replaced afterwards). It was confirmed that the catalytic asparagine (Asn) 1643 of the TSC2 was appropriately positioned to interact with the gamma-phosphate of GTP. The interface residues of the complex were defined based on a simple distance criterion, that one non-hydrogen atom of the side-chain of the residue should be within 4.5 Å of any non-hydrogen atom of the other component.

## Detailed differentiation protocols for generating hESC-derived cell types

### In vitro differentiation of hESCs into hepatic-like cells

Differentiation of hepatocyte-like cells was performed as described previously (*Si-Tayeb et al., 2010*). Briefly, hESCs were plated on matrigel-coated plates as single cells and cultured in the presence of ROCK inhibitor for 24 hr. After reaching 80% confluency, differentiation was initiated by

adding RPMI medium supplemented with B27 (GIBCO, Life Technologies), 0.5% nonessential amino acids (Life Technologies) and Activin A (100 ng/ml, R&D Systems) for 5 days to induce definitive endoderm stage cells. Cells were further differentiated by exposure to RPMI medium supplemented with B27, 0.5% nonessential amino acids, FGF4 (10 ng/ml, R&D Systems) and BMP4 (20 ng/ml, R&D Systems) for 5 days. After an additional 5 days of cultivation in RPMI/B27 supplemented with 0.5% nonessential amino acids and HGF (20 ng/ml, Peprotech), cultures mainly consisted of immature hepatocytes. Maturation to hepatocyte-like cells was initiated by cultivation in HBM medium (Lonza, UK) supplemented with HCM SingleQuots (Lonza) and Oncostatin M (20 ng/ml, R&D Systems). Hepatocytes were probed for cell-specific markers, such as $\alpha$-fetoprotein (AFP) and hepatocyte nuclear factor 4$\alpha$ (HNF4$\alpha$).

### In vitro differentiation of hESCs into neural precursors and neurons

Differentiation of neurons using an embryoid body (EB)-based feeder-free protocol was performed as described previously (*Soldner et al., 2009*; *Chambers et al., 2009*). Briefly, hESC colonies were enzymatically removed from the MEF feeder layer using 1 mg/ml collagenase type IV. The hESC clusters were collected, rinsed, re-suspended in NPM medium (DMEM/F12 medium supplemented with B-27, 1 mM L-Glutamine, 1% non-essential amino acids, 100 U/ml penicillin/streptomycin) and cultured as free–floating clusters in a low-adherent suspension culture plates; culture medium was replenished every 3–4 days. In the first 4 days of differentiation, hESC clusters were cultured in NPM medium in the presence of 10 μM SB431542 (Stemgent; inhibitor of activin and TGFβ signaling), 20 ng/ml FGF2 (R&D Systems) and 500 ng/ml noggin (Peprotech). For the next 10 days, the clusters were cultured in NPM medium supplemented with 20 ng/ml FGF2 and 500 ng/ml noggin. The clusters were finally cultured for further 7 days in NPM medium supplemented with 20 ng/ml FGF2 only. After 3 weeks of culturing, the clusters were highly enriched for neural precursors (NPs) that could be further expanded and scaled up. For this purpose, clusters were collected, briefly treated with accutase (Life Technologies) and dissociated by gently pipetting. Cells were then plated in high density and cultured in NPM medium supplemented with 20 ng/ml FGF2 and 20 ng/ml EGF (R&D Systems) on plates pre-coated with 10 μg/ml poly-D-lysine and 4 μg/ml laminin for 1–2 weeks for expansion. The culture medium was replenished every 3–4 days. For further neuronal differentiation, NPs were passaged and plated in low density on poly-D-lysine/laminin–coated plates and cultured in DMEM/F12 medium supplemented with N-2 (Life Technologies), B-27 (Life Technologies), 1 mM L-Glutamine, 1% non-essential amino acid, 100 U/ml penicillin/streptomycin with no mitogens for 2-3 weeks. Mature neurons were probed for cell-specific markers, such as neuronal class III β-tubulin (Tuj1).

### In vitro differentiation of hESCs into fibroblast-like cells

hESCs were differentiated into fibroblast-like cells as described previously (*Xu et al., 2004*). Briefly, hESC colonies were harvested using 1 mg/ml collagenase type IV, separated from the MEF feeder cells by gravity, gently triturated and cultured for 10 days in low-adherent suspension culture plates in fibroblast medium containing 20% FBS to form EBs. EBs were subsequently plated onto adherent tissue culture dishes and passaged according to primary fibroblast protocols using trypsin for at least four passages.

## Quantification and statistical analysis

Quantification of confocal images was carried out by two separate techniques. First, the percentage of cells with punctate mTOR or TSC2 (as a proxy for membrane localization of these proteins) was blindly scored. >300 cells were counted per slide and quantification is based on at least three independent experiments unless otherwise stated. Second, confocal images were collected and the co-localization plug-in in ImageJ was used to measure the co-localization between mTOR or TSC2 and the lysosomal protein Lamp1. A constant threshold was applied to all the images in the z-stack, and for every image within each experiment. Following application of the co-localization plug-in, all channels were projected (max) and quantified using Analyse particle plugin (particles 5 pixels and above were included). The data were expressed as a percentage of mTOR or TSC2 that co-localized with Lamp1. Quantification was carried out on 20–40 cells per condition from at three independent experiments. Quantification of immunoblots was carried out using ImageJ software (NIH). Two-

tailed, unpaired Student's t-tests were carried out on experimental data from at least three individual experiments.

## Acknowledgements

We are grateful for funding from BBSRC, MRC, British Skin Foundation and NIHR (VIK) and the Association for International Cancer Research (Career Development Fellowship [No.06-914/915] (to A.R. T.); Addgene, David Kwiatkowski (Harvard University, Boston) and Mark Nellist (Erasmus MC, Rotterdam) and Brendan Manning (Harvard School of Public Health, Boston) for valuable reagents; Karen Vousden (Beatson Institute for Cancer Research, Glasgow) for providing expertise in metabolomics; Manuele Rebsamen and Giulio Superti-Furga (CeMM, Vienna) for technical advice on SLC38A9 immunoblotting; S Drain, L Xie and MA Cohen for technical assistance. SS and VIK are Former Fellows of Hughes Hall, University of Cambridge, UK. Authors declare no conflict of interest.

## Additional information

### Funding

| Funder | Author |
| --- | --- |
| Biotechnology and Biological Sciences Research Council | Viktor I Korolchuk |
| Medical Research Council | Viktor I Korolchuk |
| British Skin Foundation | Viktor I Korolchuk |
| National Institute for Health Research | Viktor I Korolchuk |

The funders had no role in study design, data collection and interpretation, or the decision to submit the work for publication.

### Author contributions

Bernadette Carroll, Final approval of the version to be published, Project development and coordination, Experimental design, Performance, Evaluation, Figure preparation, Manuscript writing and review, Conception and design, Acquisition of data, Analysis and interpretation of data, Drafting or revising the article; Dorothea Maetzel, Final approval of the version to be published, Experimental design, Performance, Acquisition of data, Analysis and interpretation of data, Drafting or revising the article; Oliver DK Maddocks, Final approval of the version to be published, Experimental design, Performance and evaluation, Conception and design, Acquisition of data, Analysis and interpretation of data, Drafting or revising the article; Gisela Otten, Final approval of the version to be published; Experimental design, Performance, Acquisition of data, Analysis and interpretation of data, Drafting or revising the article; Matthew Ratcliff, Final approval of the version to be published; Experimental performance, Acquisition of data, Analysis and interpretation of data, Drafting or revising the article; Graham R Smith, Final approval of the version to be published; Bioinformatics, Acquisition of data, Analysis and interpretation of data, Drafting or revising the article; Elaine A Dunlop, Final approval of the version to be published; Experimental performance, Acquisition of data, Analysis and interpretation of data, Contributed unpublished essential data or reagents; João F Passos, Final approval of the version to be published; Evaluation and manuscript review, Acquisition of data, Drafting or revising the article, Contributed unpublished essential data or reagents; Owen R Davies, Final approval of the version to be published; Experimental design, Performance, Manuscript review, Conception and design, Acquisition of data, Analysis and interpretation of data, Drafting or revising the article; Rudolf Jaenisch, Final approval of the version to be published; Evaluation and manuscript review, Analysis and interpretation of data, Drafting or revising the article, Contributed unpublished essential data or reagents; Andrew R Tee, Final approval of the version to be published; Experimental design, Performance, Evaluation, Figure preparation, Review, Acquisition of data, Analysis and interpretation of data, Drafting or revising the article, Contributed unpublished essential data or reagents; Sovan Sarkar, Final approval of the version to be published; Experimental design,

Performance, Evaluation, Figure preparation, Review, Conception and design, Acquisition of data, Analysis and interpretation of data, Drafting or revising the article; Viktor I Korolchuk, Final approval of the version to be published; Project development and coordination, Experimental design, Manuscript writing and review, Conception and design, Acquisition of data, Analysis and interpretation of data, Drafting or revising the article

## Author ORCIDs
Elaine A Dunlop ⬤ http://orcid.org/0000-0002-9209-7561

## Decision letter and Author response
Decision letter https://doi.org/10.7554/eLife.11058.sa1
Author response https://doi.org/10.7554/eLife.11058.sa2

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
