## [Decision Letter]

Thank you for submitting your work entitled "Central role for arginine in the mechanism of nutrient sensing by mTORC1" for peer review at *eLife*. Your submission has been favorably evaluated by Sean Morrison (Senior Editor) and three reviewers, one of whom is a member of our Board of Reviewing Editors.

The reviewers have discussed the reviews with one another and the Reviewing editor has drafted this decision to help you prepare a revised submission.

The authors report that, unlike other amino acids such as leucine, Arg is important for growth factor signaling to mTORC1 by negatively regulating the lysosomal translocation of the TSC complex. The depletion of arginine (but not leucine) induced the interaction between TSC2 and Rheb. The authors further suggest that the TSC complex inhibits mTORC1 activity not only by stimulating Rheb GTPase activity as a GAP, but also by sequestering the active Rheb from mTORC1 on the lysosomes. On the other hand, involvements of Rag, SLC38A9, and mTOR in the TSC2 translocation are minor. Finally, Arg is important for a wide range of cell types including hESCs. mTORC1 dependence on Leu is observed in differentiated cells but not in ESCs. Although there is no discussion of how TSC2 is sensing arginine, overall the manuscript represents a significant advance in this important field.

Essential revisions:

1) The authors propose the model where arginine itself but not its metabolites blocks lysosomal localization of TSC2 likely though its inhibitory effect on the interaction between TSC2 and active Rheb. Is this due to a direct effect of arginine on TSC2 or Rheb, which interferes the interaction between TSC2 and active Rheb (GTP-bound form)? The effect of arginine should be studied in vitro using the immunoprecipitated TSC complex and the purified GTP-bound form of Rheb.

2) Since both the TSC complex and mTORC1 directly interact with GTP-loaded Rheb in vivo, the model that TSC inhibits GTP-loaded Rheb by steric hindrance should be directly tested in cells in the presence or absence of arginine. Do both wild type and the mutant TSC2 sufficiently block the interaction between active Rheb and mTORC1 under arginine-deficient conditions?

3) It is not convincing if this pathway is indeed independent of or parallel to the SLC38A9 pathway. In Figure 1—figure supplement 3F, there is essentially no difference between siScr and siSLC38A9, but the knockdown of SLC38A9 is not supported by any data. Since these data suggest SLC38A9 does not affect mTORC1 activity, it is critical to demonstrate its knockdown efficiency. Is it also possible to include some positive controls to show that the function of SLC38A9 is suppressed? Additionally, the authors need to provide an explanation for the difference between the current results and published results by Sabatini's and Superti-Furga's groups.

4) The biochemical data showing that Arg starvation causes TSC2 translocation to the lysosome (Figure 2E) is important to support the microscopy data. However, it is unclear how specifically lysosomes are enriched in the "lysosome" fraction. In Figure 2—figure supplement 2H, only peroxisome and plasma membrane markers are tested. At least major organelles such as the ER and mitochondria should be included.

---

## [Author Response]

The authors report that, unlike other amino acids such as leucine, Arg is important for growth factor signaling to mTORC1 by negatively regulating the lysosomal translocation of the TSC complex. The depletion of arginine (but not leucine) induced the interaction between TSC2 and Rheb. The authors further suggest that the TSC complex inhibits mTORC1 activity not only by stimulating Rheb GTPase activity as a GAP, but also by sequestering the active Rheb from mTORC1 on the lysosomes. On the other hand, involvements of Rag, SLC38A9, and mTOR in the TSC2 translocation are minor. Finally, Arg is important for a wide range of cell types including hESCs. mTORC1 dependence on Leu is observed in differentiated cells but not in ESCs. Although there is no discussion of how TSC2 is sensing arginine, overall the manuscript represents a significant advance in this important field.

The comments allowed us to improve the paper. Most importantly, the suggested experiments strengthened our conclusions about the role of arginine in disrupting the interaction between TSC2 and Rheb. Specifically, the new in vitro experiments performed with recombinant Rheb and immunoprecipitated TSC2 demonstrated an effect of arginine on binding and nucleotide loading of Rheb. These experiments do not demonstrate a direct effect of arginine on Rheb-TSC2 interaction (since a potential arginine sensor can be co-purified with TSC2) however they strongly support the role of arginine in regulating mTORC1 via this signalling axis.

Essential revisions: 1) The authors propose the model where arginine itself but not its metabolites blocks lysosomal localization of TSC2 likely though its inhibitory effect on the interaction between TSC2 and active Rheb. Is this due to a direct effect of arginine on TSC2 or Rheb, which interferes the interaction between TSC2 and active Rheb (GTP-bound form)? The effect of arginine should be studied in vitro using the immunoprecipitated TSC complex and the purified GTP-bound form of Rheb.

We thoroughly agree that the experiment suggested here will help further elucidate the specific role of arginine on TSC2-Rheb interactions. In collaboration with the lab of Owen Davies at Newcastle University we were able to purify recombinant GST-Rheb. This was loaded with non-hydrolysable GTP and incubated with TSC2 complexes immunoprecipitated from HEK293T cells, in the presence or absence of arginine (100µM). We found that indeed, the presence of arginine perturbed the interaction of Rheb and TSC2 (Figure 4E). Furthermore, GAP assays show that in the presence of arginine (100µM) there is a small perturbation of TSC2-dependent GTP hydrolysis of Rheb (Figure 4F,G).

These new data do not exclude an indirect effect of arginine by acting upon a putative arginine sensor co-immunoprecipitated with TSC2 (this is made clear in the text), however they strongly support our conclusions about the role of arginine in controlling TSC2-Rheb interaction.

2) Since both the TSC complex and mTORC1 directly interact with GTP-loaded Rheb in vivo, the model that TSC inhibits GTP-loaded Rheb by steric hindrance should be directly tested in cells in the presence or absence of arginine. Do both wild type and the mutant TSC2 sufficiently block the interaction between active Rheb and mTORC1 under arginine-deficient conditions?

We performed the experiments addressing this point where cells were transfected with TSC2 WT and GAP-deficient mutants, R1743Q and L1624P, and immunoprecipitations of GST-Rheb were carried out to monitor the effect of these mutants on its binding with mTOR. We find that R1743Q is as effective as WT TSC2 at disrupting the interaction of Rheb and mTOR while the additional TSC2 L1624P has a lesser effect but is still able to cause some dissociation between mTOR and Rheb (Figure 4—figure supplement 1J)

3) It is not convincing if this pathway is indeed independent of or parallel to the SLC38A9 pathway. In Figure 1—figure supplement 3F, there is essentially no difference between siScr and siSLC38A9, but the knockdown of SLC38A9 is not supported by any data. Since these data suggest SLC38A9 does not affect mTORC1 activity, it is critical to demonstrate its knockdown efficiency. Is it also possible to include some positive controls to show that the function of SLC38A9 is suppressed? Additionally, the authors need to provide an explanation for the difference between the current results and published results by Sabatini's and Superti-Furga's groups.

We have now included Western blot data to confirm the knock-down of SLC38A9 (Figure 1—figure supplement 3G). Apologies but we were unable to find a suitable positive control for the knock down of SLC38A9 in the literature. The efficiency of knock-down is similar to that reported by Sabatini and Superti-Furga groups and we confirm their previous findings that knock-down of SLC38A9 leads to a significant delay in recovery following amino acid (specifically arginine) starvation and re-addition (Figure 1—figure supplement 3G panel 1, left hand side). Both groups show SLC38A9 as a critical component of the Ragulator-Rag amino acid sensing pathway and Guan’s group have shown similar kinetics of mTORC1 perturbation in Rag knock-out MEFs characterised by a significant delay and dampening in recovery of mTORC1 activity following re-addition of amino acids (Jewell et al, Science, 2015). It is not clear from either Sabatini or Superti-Furga reports whether basal levels of mTORC1 are affected by SLC38A9. Our data would suggest mTORC1 is still active in basal conditions in SLC38A9 depleted cells (similar to Rag KO MEFs (Jewell et al, Science, 2015)). Furthermore, in support of both previous papers, we find that mTORC1 is effectively inhibited upon amino acid removal, i.e. there is no defect in sensing loss of amino acids.

We believe our data is in line with previously published reports and there is little discrepancy.

4) The biochemical data showing that Arg starvation causes TSC2 translocation to the lysosome (Figure 2E) is important to support the microscopy data. However, it is unclear how specifically lysosomes are enriched in the "lysosome" fraction. In Figure 2 2H, only peroxisome and plasma membrane markers are tested. At least major organelles such as the ER and mitochondria should be included.

These have now been included (Figure 2—figure supplement 2H). There is significant enrichment of lysosomes in the preps but also some contamination from other cellular membrane compartments including plasma membrane, Golgi and mitochondria. We make it clear in the text that these preparations are enriched in lysosomes rather than pure lysosomal fractions. Please note that the level of contamination by organelles other than lysosomes does not correlate with the levels of TSC2 in different starvation conditions.